# Inference of Reliability Analysis for Type II Half Logistic Weibull Distribution with Application of Bladder Cancer

**Rania A. H. Mohamed** [1,†]**, Ahlam H. Tolba** [2,*,†]**, Ehab M. Almetwally** [3,4,*,†] **and Dina A. Ramadan** [2,†]

1   Department of Statistics, Mathematics and Insurance, Faculty of Commerce, Port Said University, Port Fouad 42526, Egypt
2   Department of Mathematics, Faculty of Science, Mansoura University, Mansoura 33516, Egypt
3   Department of Statistics, Faculty of Business Administration, Delta university for Science and Technology, Gamasa 11152, Egypt
4   The Scientific Association for Studies and Applied Research, Al Manzalah 35646, Egypt
*   Correspondence: a.hamdy6@yahoo.com (A.H.T.); ehab.metwaly@deltauniv.edu.eg (E.M.A.)
†   These authors contributed equally to this work.

**Abstract:** The estimation of the unknown parameters of Type II Half Logistic Weibull (TIIHLW) distribution was analyzed in this paper. The maximum likelihood and Bayes methods are used as estimation methods. These estimators are used to estimate the fuzzy reliability function and to choose the best estimator of the fuzzy reliability function by comparing the mean square error (MSE). The simulation's results showed that fuzziness is better than reality for all sample sizes, and fuzzy reliability at Bayes predicted estimates is better than the maximum likelihood technique. It produces the lowest average MSE until a sample size of $n = 50$ is obtained. A simulated data set is applied to diagnose the performance of the two techniques applied here. A real data set is used as a practice for the model discussed and developed the maximum likelihood estimate alternative model of TIIHLW as Topp Leone inverted Kumaraswamy, modified Kies inverted Topp–Leone, Kumaraswamy Weibull–Weibull, Marshall–Olkin alpha power inverse Weibull, and odd Weibull inverted Topp–Leone. We conclude that the TIIHLW is the best distribution fit for this data.

**Keywords:** type II half logistic Weibull distribution; reliability analysis; fuzzy numbers; MCMC; highest posterior density

## 1. Introduction

The random variable was adopted for describing the objective randomness of such variable, in the conventional reliability analysis. As one of the objective facts, fuzziness exists in every problem related to real life. Fuzzy reliability theory is a new branch, which combines reliability with fuzzy mathematics. It is more reasonable that some variables and constraints are considered for the fuzziness from the point of view of engineering practice. In the analysis of fuzzy reliability, for obtaining the concrete data of reliability, the shape and values of parameters of membership function are adopted as a certainty function; therefore, the self-contradictory exists. In Ref. ([1]), the following point of views are pointed out: the books on Fuzzy Set Theory often describe a membership function on a basis of Common Set Theory and its characteristic function; it is found that the method has theoretical shortcomings of processing concept and definition absolutely. Many reliability theories and models assume that all of the parameters of the life-time probability function are strong. In real-world applications, it is necessary to generalise standard statistical estimation methods for fuzzy numbers. This is owing to the fact that the parameters of probability distributions can be represented incorrectly due to human error, personal judgement, estimation, or unexpected events. The lifetime distributions' parameters are

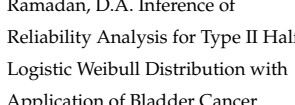



fuzzy. As a result, dealing with the function of traditional reliability may present difficulties for the system of reliability. As a result, we can deal with a larger definition of reliability than the traditional one. The degree of belonging is established by the function of a given membership, as well as the fuzzy likelihood of the vehicle or unit efficiently continuing to work for a specified period of time.

The theory of fuzzy reliability was proposed and developed by several authors; Ref. [2] discussed the Marshall-Model: Olkin's comparison of interval estimations for $P(X < Y)$, Ref. [3] introduced nonparametric reliability function estimation, [4] presented an overview of fuzzy sets, fuzzy logic, and fuzzy control systems. Furthermore, based on an exponential distribution, a fuzzy Bayesian system's reliability is assessed in [5–7], presenting fuzzy probability and statistics, Ref. [8] discussed unrepairable systems with fuzzy random lifetimes: reliability and mean time to failure, Ref. [9] studied inference on reliability in a two-parameter exponential stress–strength model. Ref. [10] showed the most effective techniques for testing fuzzy hypotheses with conflicting data, Ref. [3] studied nonparametric estimation of a reliability function, Ref. [11] presented the use of median and ranked set sampling approaches, and stress–strength reliability for exponentiated Pareto distribution can be estimated. For the inverse Rayleigh distribution, inference of a fuzzy reliability model and fuzzy reliability estimation for Frechet distribution by using simulation is presented; see [12,13], respectively. In addition, reliability estimation in Rayleigh distribution based on fuzzy lifetime data has been studied by [14,15], introducing inferences for the stress–strength reliability model's strength variable in the presence of a partially accelerated life test.

The rest of this paper is organized as follows: In Section 2, we describe the model and the formulation of fuzzy reliability. The fuzzy reliability of Type II half logistic Weibull distribution is given in Section 3. In Section 4, the ML estimators of the parameters and approximate confidence intervals are presented. We cover Bayes estimates and construction of credible intervals using the MCMC techniques in Section 5. In Section 6, we provide some simulation results in order to give an assessment of the performance of the different estimation methods. Numerical examples are presented in Section 7 for illustration.

## 2. Model and Notation

In this section, we discussed the model assumption and notation of TIIHLW distribution and fuzzy reliability analysis.

### 2.1. TIIHLW Distribution

Ref. [16] introduced modelling lifetime data from biomedical research and engineering. A variety of real data sets can properly be analyzed using the TIIHLW distribution since its density function can have various shapes (symmetric, right skewed, reversed J-shaped, and unimodal). Ref. [16] discusses the TIIHLW distribution and its features. When the cause of the failure is known or unknown, the maximum likelihood method is applied; see [17,18].

TIIHLW $(\alpha, \beta, \lambda)$ has the cumulative distribution function

$$F(x; \alpha, \beta, \lambda) = \frac{2\left[1 - e^{-\alpha x^\beta}\right]^\lambda}{1 + \left[1 - e^{-\alpha x^\beta}\right]^\lambda}; \ \alpha, \beta, \lambda > 0, x > 0, \tag{1}$$

and the probability density function is

$$f(x; \alpha, \beta, \lambda) = \frac{2\alpha \ \beta \ \lambda \ x^{\beta-1} e^{-\alpha x^\beta} \left[1 - e^{-\alpha x^\beta}\right]^{\lambda-1}}{\left[1 + \left[1 - e^{-\alpha x^\beta}\right]^\lambda\right]^2}; \ \alpha, \beta, \lambda > 0, x > 0; \tag{2}$$

the survival rate function is

$$S(x; \alpha, \beta, \lambda) = \frac{1 - \left[1 - e^{-\alpha x^\beta}\right]^\lambda}{1 + \left[1 - e^{-\alpha x^\beta}\right]^\lambda}; \; \alpha, \beta, \lambda > 0, x > 0, \tag{3}$$

and the hazard rate function is

$$h(x) = \frac{2\lambda\alpha\beta x^{\beta-1} \exp(-\alpha x^\beta)(1 - \exp(-\alpha x^\beta)^{\lambda-1}}{1 - (1 - \exp(-\alpha x^\beta))^{2\lambda}}, \tag{4}$$

where $\lambda$ and $\beta$ are shape parameters, and $\alpha$ is the scale parameter.

Figure 1 shows the probability density and hazard rate function with different shapes for the TIIHLW distribution. The behavior of the TIIHLW probability density curve may have varied shapes as seen in these illustrations. It could be skewed to the right or even to the left, or have an asymmetric or declining form, whereas the TIIHLW hazard rate curves could be falling, implying that the proposed model is a good lifetime model. As mentioned in the application section, the TIIHLW distribution has a lot of versatility when it comes to modeling skewed data, thus it is often used in domains like biology, biomedical trials, reliability, and survival studies.

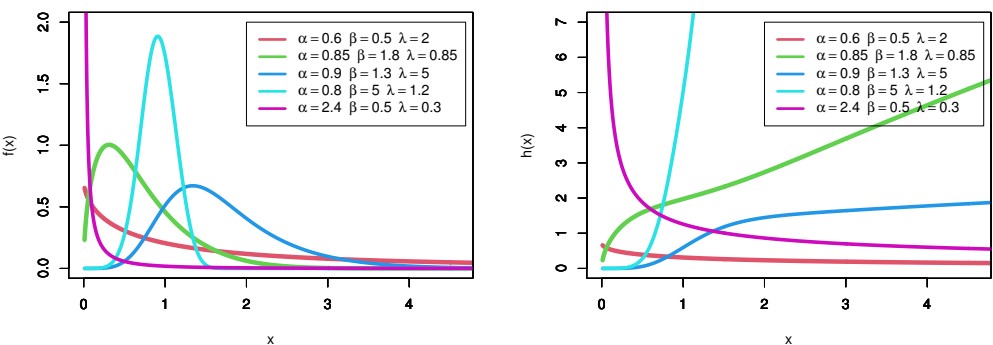

**Figure 1.** Probability density and hazard rate function with different shapes for the TIIHLW distribution.

*2.2. Fuzzy Reliability*

Let T be a continuous random variable that represents a system's failure time. The fuzzy reliability can be calculated using the fuzzy probability formula proposed by [4],

$$\tilde{R}(t) = P(T > t) = \int_t^\infty \mu(x) f(x) dx, 0 \le t \le x < \infty, \tag{5}$$

where $\mu(x)$ is a membership function that describes the degree to which each element of a given universe belongs to a fuzzy set.

Now, assume that $\mu(x)$ is

$$\mu(x) = \begin{cases} 0 & , x \le t_1 \\ \frac{x - t_1}{t_2 - t_1} & , t_1 < x < t_2, t_1 \ge 0 \\ 1 & , x \ge t_2 \end{cases} \tag{6}$$

where $0 \le t_1 \le t_2$.

For $\mu(x)$, by the computational method of the function of fuzzy numbers, the lifetime $x(\gamma)$ can be obtained corresponding to a certain value of $\gamma - Cut$, $\gamma \in [0,1]$, as follows [4]: $\mu(x) = \gamma \rightarrow \frac{x - t_1}{t_2 - t_1} = \gamma$, then

$$\begin{cases} x(\gamma) \leq t & , \gamma = 0 \\ x(\gamma) = t_1 + \gamma(t_2 - t_1) & , 0 < \gamma < 1 \\ x(\gamma) \geq t_2 & , \gamma = 1 \end{cases} \tag{7}$$

Thus, for all $\gamma$ values, fuzzy reliability values can be calculated as

$$\tilde{R}(t)_{\gamma=0} = \int_{t_1}^{t_1} f(x)\, dx = 0, \tag{8}$$

$$\tilde{R}(t)_{0<\gamma<1} = \int_{t_1}^{x(\gamma)=t_1+\gamma(t_2-t_1)} f(x)\, dx, \tag{9}$$

and

$$\tilde{R}(t)_{\gamma=1} = \int_{t_1}^{t_2} f(x)\, dx. \tag{10}$$

## 3. Fuzzy Reliability of Type II Half Logistic Weibull Distribution

We consider fuzzy reliability of type II half logistic Weibull distribution depends on the definition of fuzzy reliability. Assume that $f(x)$ in Equations (5), (8), (9), and (10) represent the pdf of Type II half logistic Weibull distribution as in Equation (2).

The fuzzy reliability definition,

$$\tilde{R}(t)_{\gamma} = \int_{t_1}^{x(\gamma)} \mu(x)\, \frac{2\alpha\,\beta\,\lambda\,x^{\beta-1}e^{-\alpha\,x^\beta}\left[1 - e^{-\alpha\,x^\beta}\right]^{\lambda-1}}{\left[1 + \left[1 - e^{-\alpha\,x^\beta}\right]^\lambda\right]^2}\, dx, \tag{11}$$

where $\mu(x)$ as in Equation (6) and $x(\gamma)$ as in Equation (7), then

$$\tilde{R}(t)_{\gamma=0} = 0, \tag{12}$$

$$\tilde{R}(t)_{0<\gamma<1} = \int_{t_1}^{t_1+\gamma(t_2-t_1)} \frac{x - t_1}{t_2 - t_1}\frac{2\alpha\,\beta\,\lambda\,x^{\beta-1}e^{-\alpha\,x^\beta}\left[1 - e^{-\alpha\,x^\beta}\right]^{\lambda-1}}{\left[1 + \left[1 - e^{-\alpha\,x^\beta}\right]^\lambda\right]^2}\, dx, \tag{13}$$

and

$$\begin{aligned} \tilde{R}(t)_{\gamma=1} &= \int_{t_1}^{t_2} \frac{2\alpha\,\beta\,\lambda\,x^{\beta-1}e^{-\alpha\,x^\beta}\left[1 - e^{-\alpha\,x^\beta}\right]^{\lambda-1}}{\left[1 + \left[1 - e^{-\alpha\,x^\beta}\right]^\lambda\right]^2}\, dx \\ &= 2\left[\left(1 + \left(1 - e^{-\alpha\,t_1^\beta}\right)^\lambda\right)^{-1} - \left(1 + \left(1 - e^{-\alpha\,t_2^\beta}\right)^\lambda\right)^{-1}\right]. \end{aligned} \tag{14}$$

## 4. Maximum Likelihood Estimation

In this section, we study type II half logistic Weibull parameters estimating problem using the maximum-likelihood estimators. Let $X$ have the Type II half logistic Weibull three-parameter PDF distribution like Equation (2).

The maximum likelihood method is given by

$$
\begin{aligned}
L(\alpha,\beta,\lambda|data) &= \prod_{i=1}^{n} f(x_i;\alpha,\beta,\lambda) \\
&= 2^n\,\alpha^n\,\beta^n\,\lambda^n \prod_{i=1}^{n} \frac{x_i^{\beta-1} e^{-\alpha\,x_i^\beta}\left[1-e^{-\alpha\,x_i^\beta}\right]^{\lambda-1}}{\left[1+\left(1-e^{-\alpha\,x_i^\beta}\right)^\lambda\right]^2}.
\end{aligned}
\tag{15}
$$

By taking log function on both sides, we obtain

$$
\begin{aligned}
\ell(\alpha,\beta,\lambda|data) &= n\,\log 2 + n\,\log\alpha + n\,\log\beta + n\,\log\lambda + (\beta-1)\sum_{i=1}^{n}\log x_i \\
&\quad -\alpha\sum_{i=1}^{n} x_i^\beta + (\lambda-1)\sum_{i=1}^{n}\log[1-e^{-\alpha\,x_i^\beta}] - 2\sum_{i=1}^{n}\log[1+\left(1-e^{-\alpha\,x_i^\beta}\right)^\lambda].
\end{aligned}
\tag{16}
$$

The type II half logistic Weibull distribution parameters $\alpha$, $\beta$ and $\lambda$ are obtained by the solution of the following equations with maximum likelihood estimator under fuzzy reliability. By differentiating the natural logarithm, $\ell(\alpha,\beta,\lambda|data)$, partially with respect to $\alpha$, $\beta$ and $\lambda$, and then equating to zero,

$$
\frac{\partial\ell}{\partial\alpha} = \frac{n}{\alpha} - \sum_{i=1}^{n} x_i^\beta + (\lambda-1)\sum_{i=1}^{n}\frac{x_i^\beta\, e^{-\alpha\,x_i^\beta}}{[1-e^{-\alpha\,x_i^\beta}]} - 2\lambda\sum_{i=1}^{n}\frac{x_i^\beta\, e^{-\alpha\,x_i^\beta}\left(1-e^{-\alpha\,x_i^\beta}\right)^{\lambda-1}}{\left[1+\left(1-e^{-\alpha\,x_i^\beta}\right)^\lambda\right]},
\tag{17}
$$

$$
\begin{aligned}
\frac{\partial\ell}{\partial\beta} &= \frac{n}{\beta} + \sum_{i=1}^{n}\log x_i + \alpha\,(\lambda-1)\sum_{i=1}^{n}\frac{x_i^\beta\,\log x_i\, e^{-\alpha\,x_i^\beta}}{[1-e^{-\alpha\,x_i^\beta}]} \\
&\quad -2\lambda\alpha\sum_{i=1}^{n}\frac{x_i^\beta\,\log x_i\, e^{-\alpha\,x_i^\beta}\left(1-e^{-\alpha\,x_i^\beta}\right)^{\lambda-1}\left[1+\left(1-e^{-\alpha\,x_i^\beta}\right)^\lambda\right]}{\left[1+\left(1-e^{-\alpha\,x_i^\beta}\right)^\lambda\right]},
\end{aligned}
\tag{18}
$$

and

$$
\frac{\partial\ell}{\partial\lambda} = \frac{n}{\lambda} + \sum_{i=1}^{n}\log[1-e^{-\alpha\,x_i^\beta}] - 2\sum_{i=1}^{n}\frac{\left(1-e^{-\alpha\,x_i^\beta}\right)^\lambda \log[1-e^{-\alpha\,x_i^\beta}]}{\left[1+\left(1-e^{-\alpha\,x_i^\beta}\right)^\lambda\right]}.
\tag{19}
$$

Since Equations (17)–(19) cannot be solved analytically, some numerical methods such as Newton's method must be employed to solve Equations (17), (18), and (19) and obtain estimates of the parameters $\alpha$, $\beta$, and $\lambda$.

The delta method is a result concerning the approximate probability distribution for a function of an asymptotically normal statistical estimator from the knowledge of the limiting variance of that estimator. Therefore, the delta method can be used to estimate the fuzzy reliability function $\tilde{R}$ of the Type II half logistic Weibull distribution, as follows:

$$
\tilde{R}(t)_{0<\gamma<1} = 2\left[\left(1+\left(1-e^{-\hat{\alpha}\,t_1^{\hat{\beta}}}\right)^{\hat{\lambda}}\right)^{-1} - \left(1+\left(1-e^{-\hat{\alpha}\,(x(\gamma))^{\hat{\beta}}}\right)^{\hat{\lambda}}\right)^{-1}\right],\quad x(\gamma)=t_1+\gamma(t_2-t_1)
\tag{20}
$$

and

$$\tilde{R}(t)_{\gamma=1} = 2\left[\left(1 + \left(1 - e^{-\hat{\alpha}\ t_1^{\hat{\beta}}}\right)^{\hat{\lambda}}\right)^{-1} - \left(1 + \left(1 - e^{-\hat{\alpha}\ t_2^{\hat{\beta}}}\right)^{\hat{\lambda}}\right)^{-1}\right], \quad (21)$$

*4.1. Confidence Intervals*

Along with the point estimator, another statistic of interest is the confidence interval estimator. The probability that the interval includes the parameter value is what we call the confidence level. Since the ML estimators of the parameters cannot be defined in analytic forms, the actual distributions of ML estimators cannot be derived.

Approximate Confidence Intervals

This subsection presents the obtaining of $100(1 - \delta)$ CI for the parameters using the asymptotic Fisher information matrix $\hat{I}(\alpha, \beta, \lambda)$, which is given by

$$\hat{I}(\alpha, \beta, \lambda) = \begin{pmatrix} -\frac{\partial^2 \ell}{\partial \alpha^2} & -\frac{\partial^2 \ell}{\partial \alpha \partial \beta} & -\frac{\partial^2 \ell}{\partial \alpha \partial \lambda} \\ -\frac{\partial^2 \ell}{\partial \beta \partial \alpha} & -\frac{\partial^2 \ell}{\partial \beta^2} & -\frac{\partial^2 \ell}{\partial \beta \partial \lambda} \\ -\frac{\partial^2 \ell}{\partial \lambda \partial a} & -\frac{\partial^2 \ell}{\partial \lambda \partial \beta} & -\frac{\partial^2 \ell}{\partial \lambda^2} \end{pmatrix}_{\downarrow\{\alpha=\hat{\alpha}, \beta=\hat{\beta}, \lambda=\hat{\lambda}\}}. \quad (22)$$

Therefore, the asymptotic variance-covariance matrix is given by

$$\hat{V} = \hat{I}^{-1}(\hat{\alpha}, \hat{\beta}, \hat{\lambda}) = \begin{pmatrix} \widehat{var(\alpha)} & cov(\alpha, \beta) & cov(\alpha, \lambda) \\ cov(\beta, \alpha) & \widehat{var(\beta)} & cov(\beta, \lambda) \\ cov(\lambda, \alpha) & cov(\lambda, \beta) & \widehat{var(\lambda)} \end{pmatrix}_{\downarrow\{\hat{\alpha}, \hat{\beta}, \hat{\lambda}\}}. \quad (23)$$

Hence, $\alpha$, $\beta$, and $\lambda$ have approximately normal distribution with mean vector $(\hat{\alpha}, \hat{\beta}, \hat{\lambda})$ and covariance matrix $\hat{I}^{-1}(\alpha, \beta, \lambda)$. Thus, the $(1 - \delta)100\%$ ACIs for $\alpha$, $\beta$, and $\lambda$ are given by

$$[\hat{\alpha} \pm Z_{\delta/2}\sqrt{\widehat{var(\alpha)}}], \quad [\hat{\beta} \pm Z_{\delta/2}\sqrt{\widehat{var(\beta)}}], \quad [\hat{\lambda} \pm Z_{\delta/2}\sqrt{\widehat{var(\lambda)}}], \quad (24)$$

where $Z_{\delta/2}$ is the percentile of the standard normal distribution with right-tail probability $\gamma/2$.

In order to compute the asymptotic CI of the fuzzy reliability function, we must first find the variance. To find approximate estimates of $\tilde{R}$ variance, the delta approach was applied. For MLE functions, the delta technique gives a general method for determining CIs [19]. It takes a function that is too complicated to calculate the variance analytically, makes a linear approximation of it, and then computes the variance of the smaller linear function that may be used for large sample inference; see [20]. We define

$$\acute{G} = \left(\frac{\partial \tilde{R}}{\partial \alpha}, \frac{\partial \tilde{R}}{\partial \beta}, \frac{\partial \tilde{R}}{\partial \lambda}\right).$$

Then, the approximate estimates of $Var(\hat{\tilde{R}})$ are given, respectively, by

$$\widehat{Var}(\hat{\tilde{R}}) \simeq \left[\acute{G}I^{-1}G\right]_{(\alpha, \beta, \lambda) = (\hat{\alpha}_{ML}, \hat{\beta}_{ML}, \hat{\lambda}_{ML})}.$$

In addition, calculate the following statistic:

$$\frac{\left(\hat{\tilde{R}} - \tilde{R}\right)}{\sqrt{\widehat{Var}(\hat{\tilde{R}})}} \sim N(0, 1);$$

these results yield the approximate CIs for $\tilde{R}$ as

$$\hat{\tilde{R}} \pm Z_{\frac{\gamma}{2}} \sqrt{\widehat{Var}(\hat{\tilde{R}})}. \tag{25}$$

## 5. Bayesian Estimation

The Bayesian approach addresses the parameters randomly and uncertainties about the parameters are represented with a joint prior distribution, established before the failed data are collected. The flexibility to incorporate prior knowledge into the analyses makes the Bayesian approach very valuable in assessing reliability since the limited availability of data is one of the primary problems of reliability analysis.

The parameters $\alpha$, $\beta$, and $\lambda$ are assumed to be independent and follow the gamma distributions. This is done accordingly

$$
\begin{aligned}
\pi_1(\alpha) &\propto \alpha^{a_1-1} e^{-b_1 \alpha} & &, \alpha > 0, a_1 > 0, b_1 > 0, \\
\pi_2(\beta) &\propto \beta^{a_2-1} e^{-b_2 \beta} & &, \beta > 0, a_2 > 0, b_2 > 0, \\
\pi_3(\lambda) &\propto \lambda^{a_3-1} e^{-b_3 \lambda} & &, \lambda > 0, a_3 > 0, b_3 > 0,
\end{aligned}
\tag{26}
$$

where the hyper-parameters $a_i$ and $b_i$, $i = 1, 2, 3$ are assumed to be known and selected to reflect the prior assumption on the unknown parameters.

The posterior distribution of the parameters $\alpha$, $\beta$ and $\lambda$ denoted by $\pi^*(\alpha, \beta, \lambda \mid data)$ up to proportionality can be obtained by combining the likelihood function Equation (15) with the prior via Bayes' theorem, and it can be written as

$$\pi^*(\alpha, \beta, \lambda \mid data) = \frac{L(\alpha, \beta, \lambda \mid data)\, \pi_1(\alpha)\, \pi_2(\beta)\, \pi_3(\lambda)}{\int\limits_0^\infty \int\limits_0^\infty \int\limits_0^\infty L(\alpha, \beta, \lambda \mid x)\, \pi_1(\alpha)\, \pi_2(\beta)\, \pi_3(\lambda)\, d\alpha\, d\beta\, d\lambda}. \tag{27}$$

The joint posterior to the proportionality can be written as

$$
\begin{aligned}
\pi^*(\alpha, \beta, \lambda \mid data) \quad \propto \quad & 2^n\, \alpha^{n+a_1-1}\, \beta^{n+a_2-1}\, \lambda^{n+a_3-1} e^{-b_1\alpha - b_2\beta - b_3\lambda} \\
& \prod_{i=1}^n \frac{x_i^{\beta-1} e^{-\alpha\, x_i^\beta} \left[1 - e^{-\alpha\, x_i^\beta}\right]^{\lambda-1}}{\left[1 + \left(1 - e^{-\alpha\, x_i^\beta}\right)^\lambda\right]^2}.
\end{aligned}
\tag{28}
$$

Now, according to Equations (12)–(14), we obtain

$$\tilde{R}(t)_{\gamma=0} = 0, \tag{29}$$

$$\tilde{R}(t)_{0<\gamma<1} = 2\left[\left(1 + \left(1 - e^{-\alpha\, t_1^\beta}\right)^\lambda\right)^{-1} - \left(1 + \left(1 - e^{-\alpha\, (x(\gamma))^\beta}\right)^\lambda\right)^{-1}\right], \quad x(\gamma) = t_1 + \gamma(t_2 - t_1), \tag{30}$$

and

$$\tilde{R}(t)_{\gamma=1} = 2\left[\left(1 + \left(1 - e^{-\alpha\, t_1^\beta}\right)^\lambda\right)^{-1} - \left(1 + \left(1 - e^{-\alpha\, t_2^\beta}\right)^\lambda\right)^{-1}\right]. \tag{31}$$

As a symmetric loss function, the squared error loss is

$$
\begin{aligned}
\tilde{R}(t)_\gamma^{BS} &= E[\tilde{R}(t)_\gamma] \\
&= \int\limits_0^\infty \tilde{R}(t)_\gamma\, \pi^*(\alpha, \beta, \lambda \mid data)\, d\alpha\, d\beta\, d\lambda.
\end{aligned}
\tag{32}
$$

Then, we obtained

$$
\tilde{R}(t)_{0<\gamma<1}^{BS} = \int_0^\infty 2^{n+1}\left[\left(1+\left(1-e^{-\alpha\, t_1^\beta}\right)^\lambda\right)^{-1} - \left(1+\left(1-e^{-\alpha\,(x(\gamma))^\beta}\right)^\lambda\right)^{-1}\right]
$$
$$
\alpha^{n+a_1-1}\,\beta^{n+a_2-1}\,\lambda^{n+a_3-1}e^{-b_1\alpha-b_2\beta-b_3\lambda}
$$
$$
\prod_{i=1}^n \frac{x_i^{\beta-1}e^{-\alpha\, x_i^\beta}\left[1-e^{-\alpha\, x_i^\beta}\right]^{\lambda-1}}{\left[1+\left(1-e^{-\alpha\, x_i^\beta}\right)^\lambda\right]^2}\,d\alpha\,d\beta\,d\lambda,
\tag{33}
$$

and

$$
\tilde{R}(t)_{\gamma=1}^{BS} = \int_0^\infty 2^{n+1}\left[\left(1+\left(1-e^{-\alpha\, t_1^\beta}\right)^\lambda\right)^{-1} - \left(1+\left(1-e^{-\alpha\, t_2^\beta}\right)^\lambda\right)^{-1}\right]
$$
$$
\alpha^{n+a_1-1}\,\beta^{n+a_2-1}\,\lambda^{n+a_3-1}e^{-b_1\alpha-b_2\beta-b_3\lambda}\prod_{i=1}^n \frac{x_i^{\beta-1}e^{-\alpha\, x_i^\beta}\left[1-e^{-\alpha\, x_i^\beta}\right]^{\lambda-1}}{\left[1+\left(1-e^{-\alpha\, x_i^\beta}\right)^\lambda\right]^2}\,d\alpha\,d\beta\,d\lambda.
\tag{34}
$$

Both the integral in Equation (28) and the normalized constant Equation (27) have no analytical solutions. Bayesian analysis should be used to evaluate the underlying model using numerical methods. Among many methods to analyze it, we will choose the Markov Chain Monte Carlo (MCMC) simulation methodology. MCMC methods can be adjusted to obtain random drawings from the posterior distribution of density in Equation (27) without having to compute the normalized constant. After that, we can use random drawings to conduct any analysis. We need visibility and model parameters. Equation (34) contains the relation on how to estimate the fuzzy reliability after estimating the unknown parameters by using the Bayesian technique.

### 5.1. Markov Chain Monte Carlo Method

For estimating complex Bayesian models, Markov chain Monte Carlo is an effective technique. The Gibbs sampling and Metropolis–Hastings algorithms are two of the most widely used Markov chain Monte Carlo methods in statistics, statistical physics, digital communications, signal processing, and machine learning, among other fields. They have attracted the attention of researchers due to their efficiency, and remarkable results have been obtained. For more information and examples using MCMC, see [21–23].

The algorithm of Metropolis–Hastings to simulate random draws from the posterior distribution $g(\theta \mid .)$:

(1) Start with initial guess $\theta^{(0)}$;
(2) Set a size of trails $M$, and the random draws;
(3) For $i = 1, ..., M$, repeat the following steps:

    (i) Set $\theta = \theta^{(i-1)}$;
    (ii) Generate a candidate $\theta^*$ from a proposal distribution $P(\theta^* \mid \theta)$;
    (iii) Evaluate the acceptance probability

$$
\eta_\theta = \min\left[1, \frac{g(\theta^* \mid .)\, P(\theta^* \mid \theta)}{g(\theta^* \mid .)\, P(\theta^* \mid \theta)}\right];
$$

    (iv) Generate a $u_1$ from a Uniform $(0,1)$ distribution.
        If $u_1 < \eta_\theta$, accept the proposal and set $\theta^{(i)} = \theta^*$, else set $\theta^{(i)} = \theta^{(i-1)}$.

Under some regularity conditions on the proposal density $P(\theta^* \mid \theta)$, the sequence of the simulated draws $\left\{\theta^{(i)}\right\}_{i=1}^{M}$ will converge to random draws that follow the posterior density $g(\theta \mid .)$. The highest posterior density (HPD) interval as the interval estimator in the Bayesian method is used to credible confidence intervals for MCMC estimates. For more information, see Turkkan and Pham-Gia [24] and Chen and Shao [25,26].

*5.2. The Highest Posterior Density*

The highest posterior density (HPD) intervals [25] discussed this technique to generate the HPD intervals of unknown parameters of the benefit distribution. In this study, samples drawn with the proposed MH algorithm should be used to generate time-lapse estimates. From the percentile tail points, for instance, a $100(1 - \gamma)\%$ HPD interval can be obtained with two points for parameters of $(\alpha, \beta, \lambda)$, from the MCMC sampling outputs. It is sometimes useful to present the posterior median to informally check for possible asymmetry in the posterior density of a parameter. According to [25,27], the BCIs of the parameters of the model of study $(\alpha, \beta, \lambda)$ can be obtained through the essential steps of the algorithm as follows: (i) Order the sample observations generated through the M–H algorithm $\tilde{\alpha}, \tilde{\beta}, \tilde{\lambda}, \tilde{R}_F$ as $(\tilde{\alpha}^{[1]} \leq \tilde{\alpha}^{[2]} \leq ... \leq \tilde{\alpha}^{[A]})$, $(\tilde{\beta}^{[1]} \leq \tilde{\beta}^{[2]} \leq ... \leq \tilde{\beta}^{[A]})$, $(\tilde{\lambda}^{[1]} \leq \tilde{\lambda}^{[2]} \leq ... \leq \tilde{\lambda}^{[A]})$ and $(\tilde{R}^{[1]} \leq \tilde{R}^{[2]} \leq ... \leq \tilde{R}^{[A]})$, where $A$ denotes the length of the generated MH algorithm.

(ii) The $100(1 - \gamma)\%$, symmetric Bayesian credible interval for $(\alpha, \beta, \lambda)$, is given by $\left(\tilde{\alpha}^{[(\gamma/2)M]}, \tilde{\alpha}^{[(1-\gamma/2)M]}\right)$, $\left(\tilde{\beta}^{[(\gamma/2)M]}, \tilde{\beta}^{[(1-\gamma/2)M]}\right)$, $\left(\tilde{\lambda}^{[(\gamma/2)M]}, \tilde{\lambda}^{[(1-\gamma/2)M]}\right)$ and $\left(\tilde{R}^{[(\gamma/2)M]}, \tilde{R}^{[(1-\gamma/2)M]}\right)$.

## 6. Simulation

The performance of the MLE and Bayesian estimation of the parameters $\alpha, \beta, \lambda$ and fuzzy reliability of the TIIHLW distribution in terms of bias, mean square errors (MSE), and confidence intervals (CI) are evaluated in this section. For $n$ sample size, we consider the values $30, 50$, and $100$. For $n$ sample size, we consider the numbers $30, 50$, and $100$. We consider the following scenarios for the parameters $\alpha, \beta, \lambda$ and the interval of the membership function $(t_1, t_2)$:

Case 1: $\alpha = 0.5, \beta = 1.2, t_1 = 0.025, t_2 = 3$ and $\lambda = 0.8$ and $1.5$;
Case 2: $\alpha = 2.2, \beta = 1.2, t_1 = 0.1, t_2 = 1$, and $\lambda = 0.8$ and $1.5$;
Case 3: $\alpha = 2.2, \beta = 3, t_1 = 0.2, t_2 = 0.8$, and $\lambda = 0.8$ and $1.5$;
Case 4: $\alpha = 1, \beta = 1.2, t_1 = 0.2, t_2 = 0.8$, and $\lambda = 0.8$ and $1.5$.

We consider replicating the process 5000 iterations. In each setting, we obtain the bias of the estimates of the corresponding MSE and CI (lower and upper). We also obtain the corresponding coverage probabilities (CP) of MLE parameters and fuzzy reliability. These results are displayed in Tables 1–4.

In Tables 1–4, we conclude these points:

1.  The MSEs decrease as the sample size increases in all of the cases;
2.  Additionally, as the sample size increases, the bias of estimates tends to zero values;
3.  Furthermore, as the sample size grows, the length of the CI of estimates tends to zero values, with the lower values of CI increasing and the upper values of CI decreasing to true values;
4.  Furthermore, in the estimation of fuzzy reliability, when the value of $\gamma$ increases, the measures of performance are improved.
5.  These results indicate that the MLE and Bayesian estimation methods of the parameters $\alpha, \beta, \lambda$, and fuzzy reliability are asymptotically unbiased and consistent.
6.  As expected, the estimates by using the Bayesian estimation method perform better than those by using the MLE method in terms of measures of performance.
7.  When $\lambda$ values increase, then the measures of parameters get better for $\alpha$ and $\beta$ where MSE decreases while $\lambda$ does not get better where MSE increases.

8. When $\alpha$ values increase, then the measures of parameters do not get better for $\alpha$ and $\beta$ where MSE increases, while $\lambda$ does get better where MSE decreases.

9. When $\beta$ values increase, then the measures of parameters do not get better for all parameters $\alpha$, $\beta$, and $\lambda$ where MSE increases.

**Table 1.** Bias, MSE, and CI for fuzzy reliability and parameters of model in case 1.

| | | | | $\alpha = 0.5, \beta = 1.2, t_1 = 0.025, t_2 = 3$ | | | | | | | | |
|---|---|---|---|---|---|---|---|---|---|---|---|---|
| | | | | **MLE** | | | | | **Bayesian** | | | |
| $\lambda$ | $n$ | $\gamma$ | | **Bias** | **MSE** | **Lower** | **Upper** | **CP** | **Bias** | **MSE** | **Lower** | **Upper** |
| 0.8 | 30 | | $\alpha$ | 0.0836 | 0.2242 | 0.0331 | 1.4986 | 95.27% | 0.0915 | 0.0453 | 0.2536 | 0.9356 |
| | | | $\beta$ | 0.2309 | 0.3995 | 0.2758 | 2.5861 | 97.30% | 0.0280 | 0.0250 | 0.9521 | 1.5298 |
| | | | $\lambda$ | 0.1921 | 0.5406 | 0.0401 | 2.3856 | 95.27% | 0.1144 | 0.0607 | 0.5478 | 1.3713 |
| | | 0.25 | $\tilde{R}$ | −0.0048 | 0.0045 | 0.3891 | 0.6527 | 97.30% | 0.0013 | 0.0034 | 0.4356 | 0.6584 |
| | | 0.55 | $\tilde{R}$ | 0.0040 | 0.0034 | 0.6571 | 0.8842 | 95.27% | 0.0175 | 0.0028 | 0.6930 | 0.8877 |
| | | 0.9 | $\tilde{R}$ | 0.0058 | 0.0019 | 0.8035 | 0.9723 | 97.30% | 0.0169 | 0.0016 | 0.8177 | 0.9634 |
| | 50 | | $\alpha$ | 0.0601 | 0.1548 | 0.1203 | 1.3229 | 94.80% | 0.0777 | 0.0415 | 0.2720 | 0.9685 |
| | | | $\beta$ | 0.1636 | 0.2608 | 0.4144 | 2.3128 | 97.20% | 0.0254 | 0.0266 | 0.9030 | 1.5390 |
| | | | $\lambda$ | 0.1243 | 0.3210 | 0.0616 | 2.0088 | 94.80% | 0.1017 | 0.0581 | 0.5369 | 1.3373 |
| | | 0.25 | $\tilde{R}$ | −0.0011 | 0.0030 | 0.4175 | 0.6317 | 97.20% | 0.0019 | 0.0023 | 0.4417 | 0.6337 |
| | | 0.55 | $\tilde{R}$ | 0.0035 | 0.0023 | 0.6773 | 0.8630 | 94.80% | 0.0158 | 0.0020 | 0.6982 | 0.8618 |
| | | 0.9 | $\tilde{R}$ | 0.0047 | 0.0013 | 0.8179 | 0.9557 | 97.20% | 0.0158 | 0.0012 | 0.8262 | 0.9473 |
| | 100 | | $\alpha$ | 0.0171 | 0.0961 | 0.1590 | 1.1245 | 95.60% | 0.0605 | 0.0320 | 0.2735 | 0.8928 |
| | | | $\beta$ | 0.1544 | 0.2007 | 0.5294 | 2.1795 | 95.60% | 0.0317 | 0.0283 | 0.9654 | 1.6195 |
| | | | $\lambda$ | 0.0438 | 0.1655 | 0.0950 | 1.6372 | 95.60% | 0.0820 | 0.0488 | 0.5607 | 1.2195 |
| | | 0.25 | $\tilde{R}$ | −0.0013 | 0.0020 | 0.4379 | 0.6111 | 95.60% | 0.0019 | 0.0015 | 0.4585 | 0.6096 |
| | | 0.55 | $\tilde{R}$ | 0.0026 | 0.0014 | 0.6953 | 0.8432 | 95.60% | 0.0148 | 0.0014 | 0.7178 | 0.8479 |
| | | 0.9 | $\tilde{R}$ | 0.0043 | 0.0008 | 0.8324 | 0.9404 | 95.60% | 0.0153 | 0.0009 | 0.8480 | 0.9462 |
| 1.5 | 30 | | $\alpha$ | 0.0442 | 0.1455 | 0.0199 | 1.2875 | 95.00% | 0.0589 | 0.0245 | 0.3088 | 0.8259 |
| | | | $\beta$ | 0.1972 | 0.2882 | 0.4176 | 2.3768 | 96.00% | 0.0372 | 0.0222 | 0.9792 | 1.5240 |
| | | | $\lambda$ | 0.2400 | 1.1754 | 0.0334 | 3.8143 | 95.00% | 0.1241 | 0.0784 | 1.1328 | 2.0946 |
| | | 0.25 | $\tilde{R}$ | −0.0066 | 0.0043 | 0.1534 | 0.4096 | 96.00% | −0.0004 | 0.0040 | 0.1655 | 0.3996 |
| | | 0.55 | $\tilde{R}$ | 0.0052 | 0.0052 | 0.4997 | 0.7807 | 95.00% | 0.0187 | 0.0051 | 0.5142 | 0.7812 |
| | | 0.9 | $\tilde{R}$ | 0.0107 | 0.0030 | 0.7448 | 0.9570 | 96.00% | 0.0189 | 0.0028 | 0.7607 | 0.9507 |
| | 50 | | $\alpha$ | 0.0389 | 0.0837 | 0.0235 | 1.1013 | 96.80% | 0.0347 | 0.0141 | 0.3268 | 0.7495 |
| | | | $\beta$ | 0.0931 | 0.1433 | 0.5733 | 2.0129 | 93.60% | 0.0299 | 0.0167 | 0.9924 | 1.4809 |
| | | | $\lambda$ | 0.1704 | 0.6118 | 0.1727 | 3.1681 | 96.80% | 0.0908 | 0.0666 | 1.1519 | 2.0463 |
| | | 0.25 | $\tilde{R}$ | −0.0030 | 0.0028 | 0.1810 | 0.3891 | 93.60% | −0.0040 | 0.0026 | 0.1880 | 0.3857 |
| | | 0.55 | $\tilde{R}$ | 0.0018 | 0.0030 | 0.5292 | 0.7443 | 96.80% | 0.0097 | 0.0029 | 0.5301 | 0.7391 |
| | | 0.9 | $\tilde{R}$ | 0.0041 | 0.0017 | 0.7647 | 0.9241 | 93.60% | 0.0121 | 0.0016 | 0.7788 | 0.9247 |
| | 100 | | $\alpha$ | −0.0252 | 0.0394 | 0.0884 | 0.8612 | 96.00% | 0.0289 | 0.0125 | 0.3449 | 0.7635 |
| | | | $\beta$ | 0.1047 | 0.0835 | 0.7763 | 1.8331 | 95.60% | 0.0229 | 0.0153 | 1.0045 | 1.4772 |
| | | | $\lambda$ | −0.0239 | 0.2357 | 0.5247 | 2.4276 | 96.00% | 0.0896 | 0.0735 | 1.1440 | 2.0904 |
| | | 0.25 | $\tilde{R}$ | −0.0033 | 0.0013 | 0.2149 | 0.3548 | 95.60% | −0.0061 | 0.0013 | 0.2228 | 0.3565 |
| | | 0.55 | $\tilde{R}$ | −0.0025 | 0.0015 | 0.5576 | 0.7075 | 96.00% | 0.0061 | 0.0014 | 0.5693 | 0.7159 |
| | | 0.9 | $\tilde{R}$ | 0.0009 | 0.0009 | 0.7836 | 0.8986 | 95.60% | 0.0092 | 0.0009 | 0.7878 | 0.8993 |

**Table 2.** Bias, MSE, and CI for fuzzy reliability and parameters of the model in case 2.

| | | | | MLE | | | | | Bayesian | | | |
|---|---|---|---|---|---|---|---|---|---|---|---|---|
| $\lambda$ | $n$ | $\gamma$ | | Bias | MSE | Lower | Upper | CP | Bias | MSE | Lower | Upper |
| | | | | | | | $\alpha = 2.2, \beta = 1.2, t_1 = 0.1, t_2 = 1$ | | | | | |
| 0.8 | 30 | | $\alpha$ | 0.3477 | 1.0233 | 0.6840 | 4.4114 | 97.60% | 0.1850 | 0.1868 | 1.7551 | 3.2432 |
| | | | $\beta$ | 0.3663 | 0.7323 | 0.0490 | 3.0837 | 97.20% | 0.0596 | 0.0281 | 0.9554 | 1.5523 |
| | | | $\lambda$ | 0.1315 | 0.4751 | 0.0396 | 2.2591 | 97.60% | 0.0869 | 0.0444 | 0.5623 | 1.2777 |
| | | 0.25 | $\tilde{R}$ | 0.0041 | 0.0019 | 0.2722 | 0.4409 | 97.20% | 0.0184 | 0.0011 | 0.3216 | 0.4307 |
| | | 0.55 | $\tilde{R}$ | 0.0103 | 0.0038 | 0.4189 | 0.6572 | 97.60% | 0.0369 | 0.0038 | 0.4733 | 0.6657 |
| | | 0.9 | $\tilde{R}$ | 0.0113 | 0.0050 | 0.4880 | 0.7611 | 97.20% | 0.0434 | 0.0056 | 0.5399 | 0.7719 |
| | 50 | | $\alpha$ | 0.1710 | 0.4813 | 1.0518 | 3.6902 | 96.20% | 0.1299 | 0.1499 | 1.6904 | 3.0400 |
| | | | $\beta$ | 0.2055 | 0.3405 | 0.3339 | 2.4771 | 97.40% | 0.0449 | 0.0262 | 0.9207 | 1.5135 |
| | | | $\lambda$ | 0.0986 | 0.2972 | 0.1534 | 1.9506 | 96.20% | 0.0793 | 0.0398 | 0.5385 | 1.2107 |
| | | 0.25 | $\tilde{R}$ | 0.0010 | 0.0011 | 0.2877 | 0.4192 | 97.40% | 0.0159 | 0.0009 | 0.3242 | 0.4181 |
| | | 0.55 | $\tilde{R}$ | 0.0040 | 0.0022 | 0.4409 | 0.6226 | 96.20% | 0.0312 | 0.0026 | 0.4859 | 0.6397 |
| | | 0.9 | $\tilde{R}$ | 0.0048 | 0.0028 | 0.5148 | 0.7213 | 97.40% | 0.0372 | 0.0036 | 0.5549 | 0.7353 |
| | 100 | | $\alpha$ | 0.0589 | 0.1651 | 1.4701 | 3.0477 | 93.80% | 0.0783 | 0.1026 | 1.6262 | 2.8592 |
| | | | $\beta$ | 0.1236 | 0.1707 | 0.5500 | 2.0972 | 94.80% | 0.0358 | 0.0233 | 0.9437 | 1.5099 |
| | | | $\lambda$ | 0.0170 | 0.0868 | 0.2401 | 1.3940 | 93.80% | 0.0580 | 0.0352 | 0.5620 | 1.2335 |
| | | 0.25 | $\tilde{R}$ | 0.0008 | 0.0005 | 0.3081 | 0.3985 | 94.80% | 0.0135 | 0.0006 | 0.3269 | 0.4035 |
| | | 0.55 | $\tilde{R}$ | 0.0045 | 0.0012 | 0.4651 | 0.5993 | 93.80% | 0.0283 | 0.0019 | 0.4983 | 0.6224 |
| | | 0.9 | $\tilde{R}$ | 0.0061 | 0.0016 | 0.5415 | 0.6971 | 94.80% | 0.0348 | 0.0027 | 0.5733 | 0.7239 |
| 1.5 | 30 | | $\alpha$ | 0.0778 | 0.6497 | 0.7045 | 3.8511 | 95.00% | 0.1360 | 0.1495 | 1.6736 | 3.0468 |
| | | | $\beta$ | 0.4523 | 0.8849 | 0.0348 | 3.2698 | 93.90% | 0.0679 | 0.0223 | 1.0465 | 1.5581 |
| | | | $\lambda$ | −0.0360 | 0.7863 | 0.0427 | 3.2014 | 95.00% | 0.0591 | 0.0431 | 1.2037 | 1.9627 |
| | | 0.25 | $\tilde{R}$ | −0.0096 | 0.0032 | 0.2371 | 0.4573 | 93.90% | −0.0086 | 0.0019 | 0.2651 | 0.4345 |
| | | 0.55 | $\tilde{R}$ | −0.0009 | 0.0045 | 0.5094 | 0.7733 | 95.00% | 0.0083 | 0.0020 | 0.5679 | 0.7354 |
| | | 0.9 | $\tilde{R}$ | 0.0066 | 0.0034 | 0.6915 | 0.9177 | 93.90% | 0.0163 | 0.0018 | 0.7424 | 0.8951 |
| | 50 | | $\alpha$ | 0.0795 | 0.4378 | 0.9914 | 3.5676 | 95.50% | 0.1068 | 0.1144 | 1.7045 | 2.9413 |
| | | | $\beta$ | 0.2840 | 0.5679 | 0.1153 | 2.8528 | 95.10% | 0.0624 | 0.0197 | 1.0355 | 1.5186 |
| | | | $\lambda$ | 0.0311 | 0.6881 | 0.1215 | 3.4404 | 95.50% | 0.0579 | 0.0409 | 1.1519 | 1.9944 |
| | | 0.25 | $\tilde{R}$ | −0.0037 | 0.0020 | 0.2652 | 0.4411 | 95.10% | −0.0083 | 0.0012 | 0.2852 | 0.4173 |
| | | 0.55 | $\tilde{R}$ | 0.0021 | 0.0029 | 0.5397 | 0.7490 | 95.50% | 0.0082 | 0.0015 | 0.5773 | 0.7241 |
| | | 0.9 | $\tilde{R}$ | 0.0062 | 0.0021 | 0.7158 | 0.8926 | 95.10% | 0.0165 | 0.0014 | 0.7452 | 0.8773 |
| | 100 | | $\alpha$ | 0.0303 | 0.1901 | 1.3773 | 3.0832 | 95.60% | 0.0564 | 0.0661 | 1.8053 | 2.7666 |
| | | | $\beta$ | 0.1150 | 0.1554 | 0.5756 | 2.0543 | 94.50% | 0.0483 | 0.0158 | 1.0235 | 1.4706 |
| | | | $\lambda$ | 0.0641 | 0.3750 | 0.3699 | 2.7583 | 95.60% | 0.0518 | 0.0401 | 1.2120 | 2.0282 |
| | | 0.25 | $\tilde{R}$ | −0.0027 | 0.0010 | 0.2917 | 0.4166 | 94.50% | −0.0096 | 0.0008 | 0.3001 | 0.4009 |
| | | 0.55 | $\tilde{R}$ | 0.0004 | 0.0013 | 0.5716 | 0.7137 | 95.60% | 0.0045 | 0.0009 | 0.5900 | 0.7000 |
| | | 0.9 | $\tilde{R}$ | 0.0030 | 0.0010 | 0.7391 | 0.8629 | 94.50% | 0.0131 | 0.0009 | 0.7576 | 0.8631 |

**Table 3.** Bias, MSE, and CI for fuzzy reliability and parameters of models in case 3.

| | | | | MLE | | | | | Bayesian | | | |
|---|---|---|---|---|---|---|---|---|---|---|---|---|
| $\lambda$ | $n$ | $\gamma$ | | Bias | MSE | Lower | Upper | CP | Bias | MSE | Lower | Upper |
| | | | $\alpha$ | 0.4625 | 1.0897 | 0.8274 | 4.4976 | 96.20% | 0.2094 | 0.2136 | 1.6354 | 3.2315 |
| | | | $\beta$ | 0.2391 | 1.6792 | 0.7416 | 5.7366 | 96.30% | 0.0142 | 0.0105 | 2.8137 | 3.2068 |
| | 30 | | $\lambda$ | 0.3126 | 0.8681 | 0.0609 | 2.8338 | 96.20% | 0.0674 | 0.0263 | 0.6043 | 1.1440 |
| | | 0.25 | $\tilde{R}$ | 0.0010 | 0.0016 | 0.1025 | 0.2573 | 96.30% | −0.0077 | 0.0010 | 0.1055 | 0.2233 |
| | | 0.55 | $\tilde{R}$ | 0.0104 | 0.0041 | 0.3411 | 0.5886 | 96.20% | −0.0024 | 0.0016 | 0.3703 | 0.5277 |
| | | 0.9 | $\tilde{R}$ | 0.0138 | 0.0038 | 0.6118 | 0.8488 | 96.30% | 0.0089 | 0.0012 | 0.6589 | 0.7895 |
| | | | $\alpha$ | 0.2230 | 0.5556 | 1.0283 | 3.8177 | 95.70% | 0.1587 | 0.1785 | 1.6471 | 3.1347 |
| | | | $\beta$ | 0.3673 | 1.4003 | 1.1614 | 5.5732 | 96.50% | 0.0139 | 0.0101 | 2.7764 | 3.2176 |
| 0.8 | 50 | | $\lambda$ | 0.0962 | 0.2755 | 0.1155 | 1.9080 | 95.70% | 0.0515 | 0.0172 | 0.6228 | 1.0910 |
| | | 0.25 | $\tilde{R}$ | −0.0012 | 0.0008 | 0.1230 | 0.2324 | 96.50% | −0.0063 | 0.0006 | 0.1265 | 0.2197 |
| | | 0.55 | $\tilde{R}$ | 0.0015 | 0.0024 | 0.3597 | 0.5522 | 95.70% | −0.0014 | 0.0011 | 0.3883 | 0.5186 |
| | | 0.9 | $\tilde{R}$ | 0.0045 | 0.0026 | 0.6213 | 0.8207 | 96.50% | 0.0076 | 0.0011 | 0.6604 | 0.7864 |
| | | | $\alpha$ | 0.1572 | 0.2423 | 1.4425 | 3.2719 | 96.00% | 0.1155 | 0.1074 | 1.7232 | 2.9011 |
| | | | $\beta$ | 0.0916 | 0.5368 | 1.6662 | 4.5171 | 96.90% | 0.0123 | 0.0098 | 2.7795 | 3.2899 |
| | 100 | | $\lambda$ | 0.1038 | 0.1912 | 0.0707 | 1.7368 | 96.00% | 0.0404 | 0.0108 | 0.6729 | 1.0308 |
| | | 0.25 | $\tilde{R}$ | 0.0003 | 0.0004 | 0.1383 | 0.2201 | 96.90% | −0.0060 | 0.0004 | 0.1355 | 0.2056 |
| | | 0.55 | $\tilde{R}$ | 0.0041 | 0.0012 | 0.3899 | 0.5273 | 96.00% | −0.0018 | 0.0006 | 0.4064 | 0.5047 |
| | | 0.9 | $\tilde{R}$ | 0.0060 | 0.0012 | 0.6554 | 0.7897 | 96.90% | 0.0069 | 0.0007 | 0.6739 | 0.7728 |
| | | | $\alpha$ | 0.2505 | 0.8275 | 0.7356 | 4.1653 | 96.50% | 0.1352 | 0.1320 | 1.7240 | 3.0249 |
| | | | $\beta$ | 0.5053 | 2.3515 | 0.6661 | 6.3444 | 97.70% | 0.0345 | 0.0133 | 2.8154 | 3.2366 |
| | 30 | | $\lambda$ | 0.5446 | 2.9301 | 0.1378 | 5.2269 | 96.50% | 0.0875 | 0.0448 | 1.2315 | 1.9657 |
| | | 0.25 | $\tilde{R}$ | −0.0021 | 0.0006 | 0.0009 | 0.0926 | 97.70% | −0.0038 | 0.0004 | 0.0131 | 0.0811 |
| | | 0.55 | $\tilde{R}$ | −0.0070 | 0.0042 | 0.1198 | 0.3715 | 96.50% | −0.0097 | 0.0029 | 0.1458 | 0.3498 |
| | | 0.9 | $\tilde{R}$ | −0.0004 | 0.0056 | 0.4719 | 0.7654 | 97.70% | −0.0017 | 0.0032 | 0.5057 | 0.7255 |
| | | | $\alpha$ | 0.1420 | 0.4515 | 1.0540 | 3.6299 | 95.50% | 0.1084 | 0.1073 | 1.7384 | 2.9163 |
| | | | $\beta$ | 0.3371 | 1.5045 | 1.0244 | 5.6497 | 96.00% | 0.0316 | 0.0130 | 2.7169 | 3.2388 |
| 1.5 | 50 | | $\lambda$ | 0.3146 | 1.3864 | 0.1411 | 4.0396 | 95.50% | 0.0834 | 0.0397 | 1.2622 | 1.9604 |
| | | 0.25 | $\tilde{R}$ | −0.0019 | 0.0004 | 0.0091 | 0.0831 | 96.00% | −0.0047 | 0.0003 | 0.0147 | 0.0726 |
| | | 0.55 | $\tilde{R}$ | −0.0056 | 0.0025 | 0.1503 | 0.3437 | 95.50% | −0.0118 | 0.0020 | 0.1530 | 0.3180 |
| | | 0.9 | $\tilde{R}$ | −0.0010 | 0.0034 | 0.5033 | 0.7328 | 96.00% | −0.0038 | 0.0022 | 0.5230 | 0.7065 |
| | | | $\alpha$ | 0.0269 | 0.1939 | 1.3651 | 3.0887 | 95.20% | 0.0561 | 0.0657 | 1.8009 | 2.7655 |
| | | | $\beta$ | 0.2392 | 0.8517 | 1.4912 | 4.9871 | 95.10% | 0.0268 | 0.0126 | 2.7880 | 3.2628 |
| | 100 | | $\lambda$ | 0.0860 | 0.4431 | 0.2916 | 2.8805 | 95.20% | 0.0526 | 0.0281 | 1.2695 | 1.8645 |
| | | 0.25 | $\tilde{R}$ | 0.0003 | 0.0002 | 0.0209 | 0.0757 | 95.10% | −0.0036 | 0.0002 | 0.0215 | 0.0674 |
| | | 0.55 | $\tilde{R}$ | −0.0017 | 0.0013 | 0.1804 | 0.3214 | 95.20% | −0.0097 | 0.0012 | 0.1843 | 0.3126 |
| | | 0.9 | $\tilde{R}$ | −0.0022 | 0.0018 | 0.5348 | 0.6990 | 95.10% | −0.0049 | 0.0014 | 0.5380 | 0.6821 |

**Table 4.** Bias, MSE, and CI for fuzzy reliability and parameters of model in case 4.

| | | | | MLE | | | | | Bayesian | | | |
|---|---|---|---|---|---|---|---|---|---|---|---|---|
| $\lambda$ | $n$ | $\gamma$ | | Bias | MSE | Lower | Upper | CP | Bias | MSE | Lower | Upper |
| 0.8 | 30 | | $\alpha$ | 0.0432 | 0.4142 | −0.2160 | 2.3025 | 95.50% | 0.1442 | 0.1294 | 0.5982 | 1.8053 |
| | | | $\beta$ | 0.4273 | 0.7146 | 0.1970 | 3.0576 | 97.20% | 0.0648 | 0.0342 | 0.9313 | 1.5827 |
| | | | $\lambda$ | 0.0762 | 0.4601 | −0.4455 | 2.1978 | 95.50% | 0.0865 | 0.0482 | 0.5419 | 1.2648 |
| | | 0.25 | RF | 0.0014 | 0.0005 | 0.1128 | 0.2043 | 97.20% | 0.0083 | 0.0004 | 0.1303 | 0.2009 |
| | | 0.55 | RF | 0.0047 | 0.0016 | 0.2163 | 0.3721 | 95.50% | 0.0195 | 0.0015 | 0.2485 | 0.3757 |
| | | 0.9 | RF | 0.0087 | 0.0027 | 0.3044 | 0.5045 | 97.20% | 0.0287 | 0.0028 | 0.3446 | 0.5110 |
| | 50 | | $\alpha$ | 0.0567 | 0.2835 | 0.0185 | 2.0948 | 96.30% | 0.1068 | 0.0824 | 0.6537 | 1.6186 |
| | | | $\beta$ | 0.1876 | 0.2939 | 0.3902 | 2.3851 | 96.90% | 0.0345 | 0.0263 | 0.9321 | 1.5310 |
| | | | $\lambda$ | 0.1126 | 0.3397 | −0.2088 | 2.0340 | 96.30% | 0.0845 | 0.0451 | 0.5494 | 1.2083 |
| | | 0.25 | RF | 0.0013 | 0.0004 | 0.1211 | 0.1959 | 96.90% | 0.0066 | 0.0003 | 0.1366 | 0.1925 |
| | | 0.55 | RF | 0.0031 | 0.0010 | 0.2298 | 0.3553 | 96.30% | 0.0162 | 0.0010 | 0.2565 | 0.3576 |
| | | 0.9 | RF | 0.0050 | 0.0017 | 0.3213 | 0.4799 | 96.90% | 0.0244 | 0.0020 | 0.3513 | 0.4901 |
| | 100 | | $\alpha$ | 0.0255 | 0.1338 | 0.3100 | 1.7411 | 95.90% | 0.0621 | 0.0495 | 0.6924 | 1.4883 |
| | | | $\beta$ | 0.1070 | 0.1622 | 0.5456 | 2.0684 | 95.70% | 0.0342 | 0.0261 | 0.8995 | 1.5254 |
| | | | $\lambda$ | 0.0466 | 0.1217 | 0.1686 | 1.5246 | 95.90% | 0.0688 | 0.0393 | 0.5365 | 1.2094 |
| | | 0.25 | RF | 0.0007 | 0.0002 | 0.1324 | 0.1834 | 95.70% | 0.0049 | 0.0001 | 0.1430 | 0.1854 |
| | | 0.55 | RF | 0.0015 | 0.0005 | 0.2481 | 0.3338 | 95.90% | 0.0121 | 0.0005 | 0.2665 | 0.3428 |
| | | 0.9 | RF | 0.0023 | 0.0008 | 0.3438 | 0.4520 | 95.70% | 0.0187 | 0.0011 | 0.3652 | 0.4685 |
| 1.5 | 30 | | $\alpha$ | −0.0190 | 0.3250 | −0.1363 | 2.0983 | 98.10% | 0.0682 | 0.0476 | 0.7119 | 1.5046 |
| | | | $\beta$ | 0.3712 | 0.6819 | 0.1247 | 3.0177 | 94.50% | 0.0528 | 0.0231 | 0.9814 | 1.5399 |
| | | | $\lambda$ | 0.1202 | 1.1053 | −0.4280 | 3.6684 | 98.10% | 0.0782 | 0.0575 | 1.1854 | 2.0476 |
| | | 0.25 | RF | −0.0036 | 0.0007 | 0.0693 | 0.1721 | 94.50% | −0.0033 | 0.0006 | 0.0772 | 0.1683 |
| | | 0.55 | RF | −0.0048 | 0.0023 | 0.1779 | 0.3658 | 98.10% | −0.0001 | 0.0017 | 0.2009 | 0.3574 |
| | | 0.9 | RF | −0.0030 | 0.0038 | 0.3059 | 0.5482 | 94.50% | 0.0070 | 0.0026 | 0.3441 | 0.5378 |
| | 50 | | $\alpha$ | −0.0018 | 0.2499 | 0.0179 | 1.9785 | 97.30% | 0.0358 | 0.0317 | 0.7128 | 1.3734 |
| | | | $\beta$ | 0.2248 | 0.3584 | 0.3368 | 2.5128 | 94.40% | 0.0527 | 0.0192 | 1.0014 | 1.4828 |
| | | | $\lambda$ | 0.1569 | 0.9451 | −0.2245 | 3.5383 | 97.30% | 0.0761 | 0.0526 | 1.1745 | 1.9814 |
| | | 0.25 | RF | −0.0040 | 0.0005 | 0.0778 | 0.1627 | 94.40% | −0.0064 | 0.0004 | 0.0799 | 0.1550 |
| | | 0.55 | RF | −0.0054 | 0.0016 | 0.1946 | 0.3478 | 97.30% | −0.0055 | 0.0011 | 0.2057 | 0.3348 |
| | | 0.9 | RF | −0.0038 | 0.0024 | 0.3301 | 0.5224 | 94.40% | 0.0011 | 0.0016 | 0.3486 | 0.5034 |
| | 100 | | $\alpha$ | −0.0100 | 0.1471 | 0.2382 | 1.7418 | 96.30% | 0.0252 | 0.0231 | 0.7458 | 1.3359 |
| | | | $\beta$ | 0.1276 | 0.1623 | 0.5782 | 2.0769 | 96.10% | 0.0387 | 0.0162 | 1.0086 | 1.4850 |
| | | | $\lambda$ | 0.0829 | 0.5082 | 0.1945 | 2.9713 | 96.30% | 0.0707 | 0.0505 | 1.1763 | 1.9598 |
| | | 0.25 | RF | −0.0027 | 0.0002 | 0.0913 | 0.1518 | 96.10% | −0.0059 | 0.0002 | 0.0887 | 0.1450 |
| | | 0.55 | RF | −0.0038 | 0.0008 | 0.2172 | 0.3285 | 96.30% | −0.0055 | 0.0006 | 0.2204 | 0.3166 |
| | | 0.9 | RF | −0.0031 | 0.0013 | 0.3561 | 0.4979 | 96.10% | −0.0003 | 0.0009 | 0.3712 | 0.4864 |

The table header spans: $\alpha = 1, \beta = 1.2$ and $t_1 = 0.2, t_2 = 0.8$

## 7. Application of Real Data

In this section, the flexibility and potentiality of the TIIHLW distribution are examined using two real data sets. We provide an application of the TIIHLW distribution with Fuzzy reliability. The cancer data set is given by [28], which represents remission times (in months) of a random sample of 128 bladder cancer patients. In Table 5, the TIIHLW model has the highest *p*-value and the lowest distance (D) of the Kolmogorov–Smirnov (K-S) value. Bayesian estimation methods have smaller stander error (SE). Table 6 discussed the MLE estimate alternative model of TIIHLW as Topp Leone inverted Kumaraswamy (TLIK) [29], modifed Kies inverted Topp–Leone (MKITL) [30], Kumaraswamy Weibull–Weibull (KW-W) [31], Marshall–Olkin alpha power inverse Weibull (MOAPIW) [32], and Odd Weibull ITL (OWITL) [33]. Table 7 shows that the TIIHLW fits the data by the Akaike information criterion (AIC), Anderson–Darling (AD), Bayesian information criterion (BIC), and Cramér–von Mises criterion (CVM) values. Figure 2 shows the fit of the empirical CDF, histogram, and PP-plot. Figure 3 shows convergence plots of MCMC for parameter estimates of TIIHLW distribution. Figure 4 shows the estimates exist and have maximum log-likelihood value. We conclude that the TIIHLW is the best distribution fit for this data.

**Table 5.** MLE and Bayesian estimation for parameters of TIIHLW distribution.

|          |           | Estimates | SE     | Lower  | Upper  |
|----------|-----------|-----------|--------|--------|--------|
|          | $\alpha$  | 0.2008    | 0.1377 | 0.0069 | 0.4706 |
| MLE      | $\beta$   | 0.8118    | 0.1766 | 0.4656 | 1.1580 |
|          | $\lambda$ | 2.0828    | 0.8296 | 0.4568 | 3.7089 |
|          | $\alpha$  | 0.1981    | 0.0628 | 0.0844 | 0.3117 |
| Bayesian | $\beta$   | 0.8284    | 0.0943 | 0.6536 | 1.0117 |
|          | $\lambda$ | 2.0609    | 0.4079 | 1.3090 | 2.8671 |

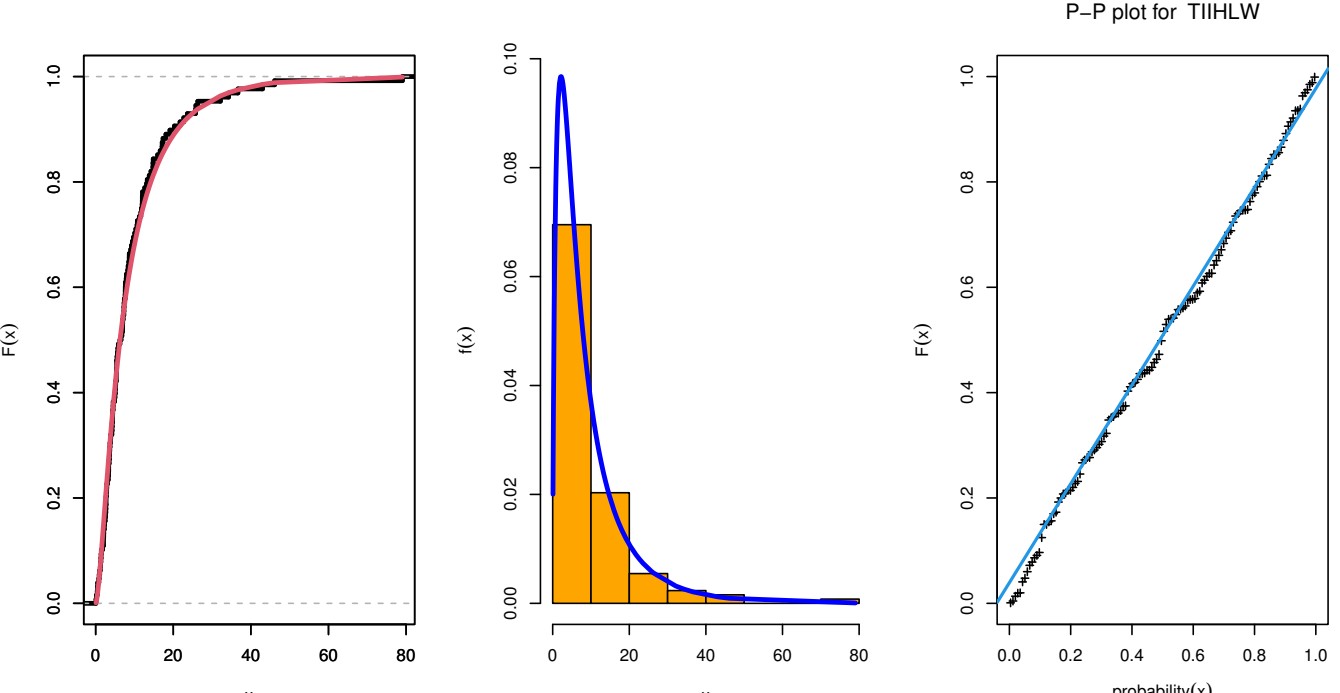

**Figure 2.** Estimated CDF and empirical CDF, estimated PDF with histogram, Q-Q plot, and P-P plot for the TIIHLW distribution.

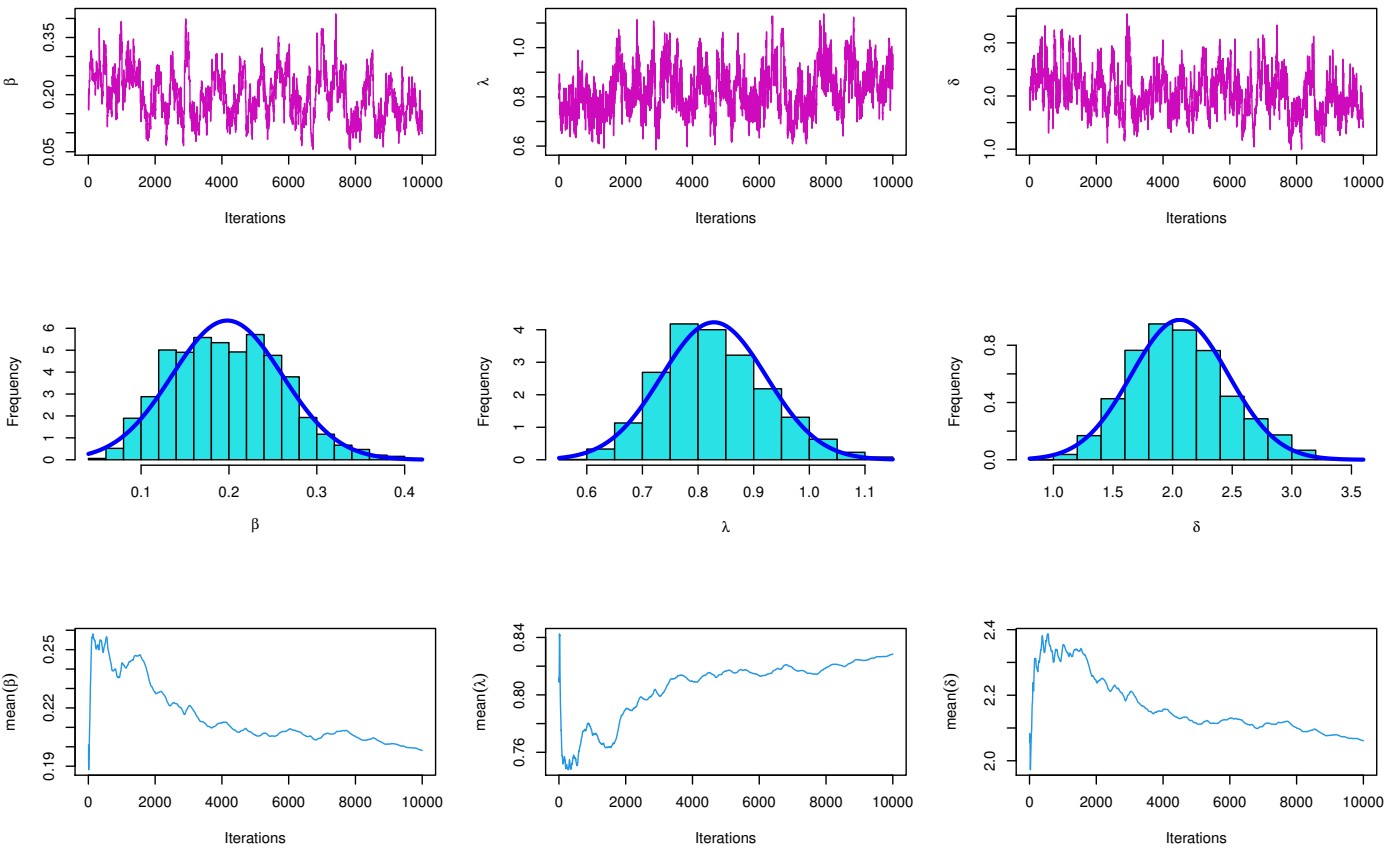

**Figure 3.** Convergence plots of MCMC for parameter estimates of TIIHLW distribution.

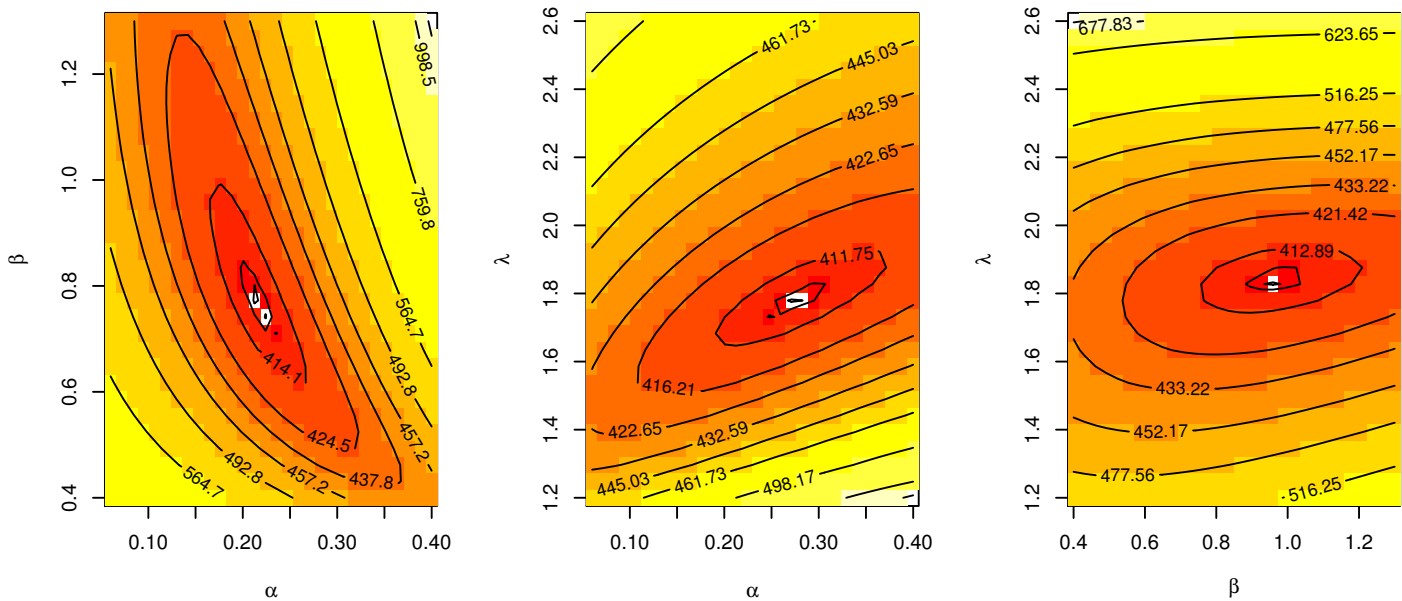

**Figure 4.** Contour plot of parameters and log-likelihood value for the TIIHLW distribution.

**Table 6.** MLE with SE for an alternative model of TIIHLW.

|   | TLIK | | MKITL | | KW-W | | MOAPIW | | OWITL | |
|---|---|---|---|---|---|---|---|---|---|---|
|   | **Estimates** | **SE** | **Estimates** | **SE** | **Estimates** | **SE** | **Estimates** | **SE** | **Estimates** | **SE** |
| $\alpha$ | 1.9018 | 0.6338 | 1.4058 | 0.1039 | 1.6434 | 0.0031 | 100.0941 | 850.8987 | 1.2686 | 0.2379 |
| $\beta$ | 9.2617 | 4.7748 | 0.4269 | 0.0196 | 0.7084 | 0.0031 | 1.7014 | 0.1306 | 0.5490 | 0.4581 |
| $\lambda$ | 0.6536 | 0.1671 | | | 4.5002 | 0.0069 | 799.9982 | 1374.6087 | 0.5814 | 0.2641 |
| $\theta$ | | | | | 0.1606 | 0.0143 | 0.0058 | 0.0028 | | |

**Table 7.** AIC, CAIC, KS test, CVM, and AD.

|   | **KS** | ***p*-Value** | **AIC** | **BIC** | **CVM** | **AD** |
|---|---|---|---|---|---|---|
| TIIHLW | 0.0404 | 0.9851 | 827.1032 | 835.6593 | 0.0396 | 0.2676 |
| TLIK | 0.0982 | 0.1697 | 855.5293 | 864.0854 | 0.3349 | 2.1509 |
| MKITL | 0.0502 | 0.9034 | 828.3362 | 834.0403 | 0.0683 | 0.4629 |
| KW-W | 0.0813 | 0.3655 | 836.8575 | 848.2656 | 0.1081 | 0.7183 |
| MOAPIW | 0.0410 | 0.9824 | 831.3024 | 842.7105 | 0.0482 | 0.3475 |
| OWITL | 0.0538 | 0.8531 | 829.9590 | 838.5150 | 0.0687 | 0.4553 |

Table 8 shows fuzzy reliability by different $\gamma$ values and different intervals of membership function. We note that, when the value of $\gamma$ increases, the fuzzy reliability that is improved tends to 1. As expected, the estimates by using the Bayesian estimation method perform better than those by using the MLE method in terms of fuzzy reliability and SE.

**Table 8.** Fuzzy reliability.

|   | **MLE** | | **Bayesian** | |
|---|---|---|---|---|
| $\gamma$ | 2, 12 | 1.5, 18 | 2, 12 | 1.5, 18 |
| 0.3 | 0.2648 | 0.4106 | 0.2723 | 0.4214 |
| 0.6 | 0.4479 | 0.6278 | 0.4584 | 0.6400 |
| 0.9 | 0.5676 | 0.7397 | 0.5785 | 0.7500 |

For Figures 5 and 6, the proposed distribution (normal) of the MH algorithm and the prior distributions of the subsequent terms of the MCMC was the same as the posterior distribution. In both cases, problems seem to have been solved: the trace plots of the MCMC samples do not show any apparent anomaly. Figure 7 shows an auto-correlation plot for fuzzy reliability estimation of TIIHLW distribution by using MCMC results. In the graph, there is a vertical line corresponding to each lag. The height of each spike shows the value of the auto-correlation function for the lag. In this figure, the spikes are statistically significant for lags up to 40. This means that the fuzzy reliability estimation of TIIHLW distribution by using MCMC results are highly correlated with each other.

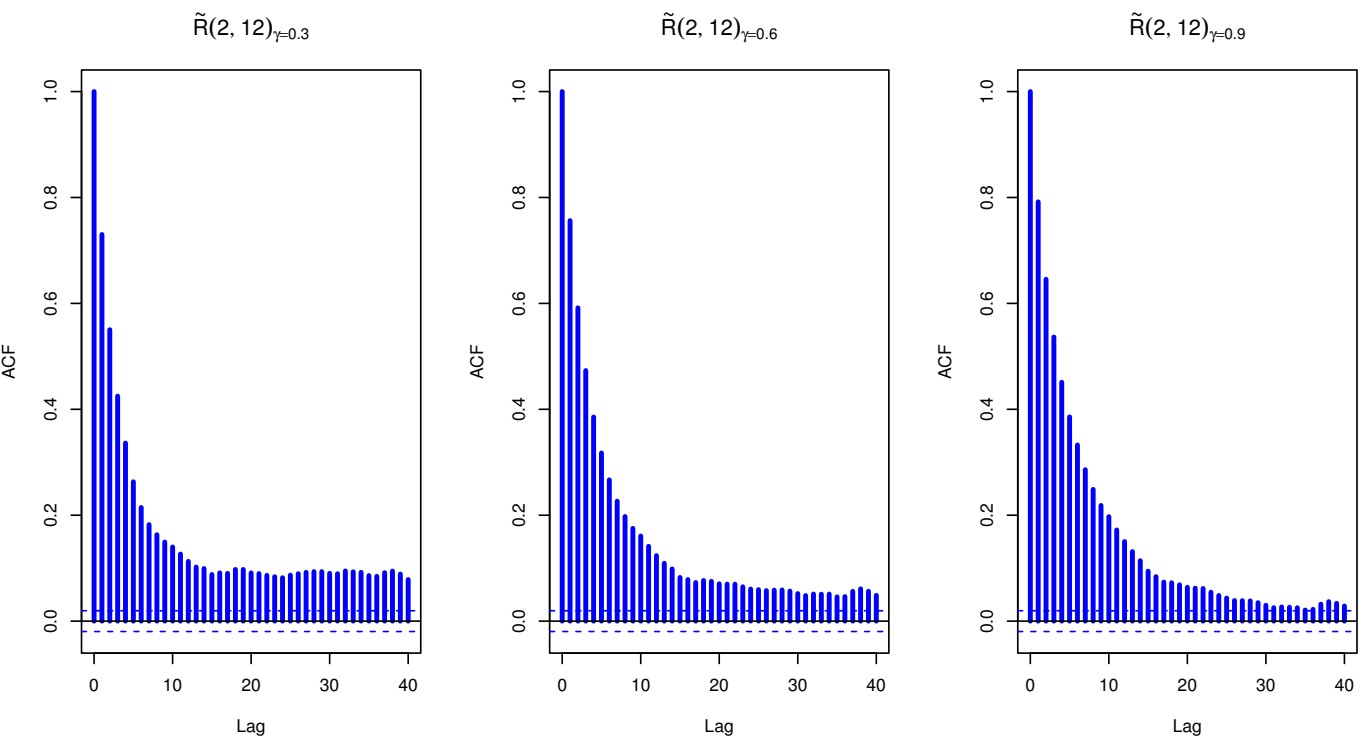

**Figure 5.** The auto-correlation plot of MCMC for fuzzy reliability $\tilde{R}$ of TIIHLW distribution where lower is 2 and upper is 12.

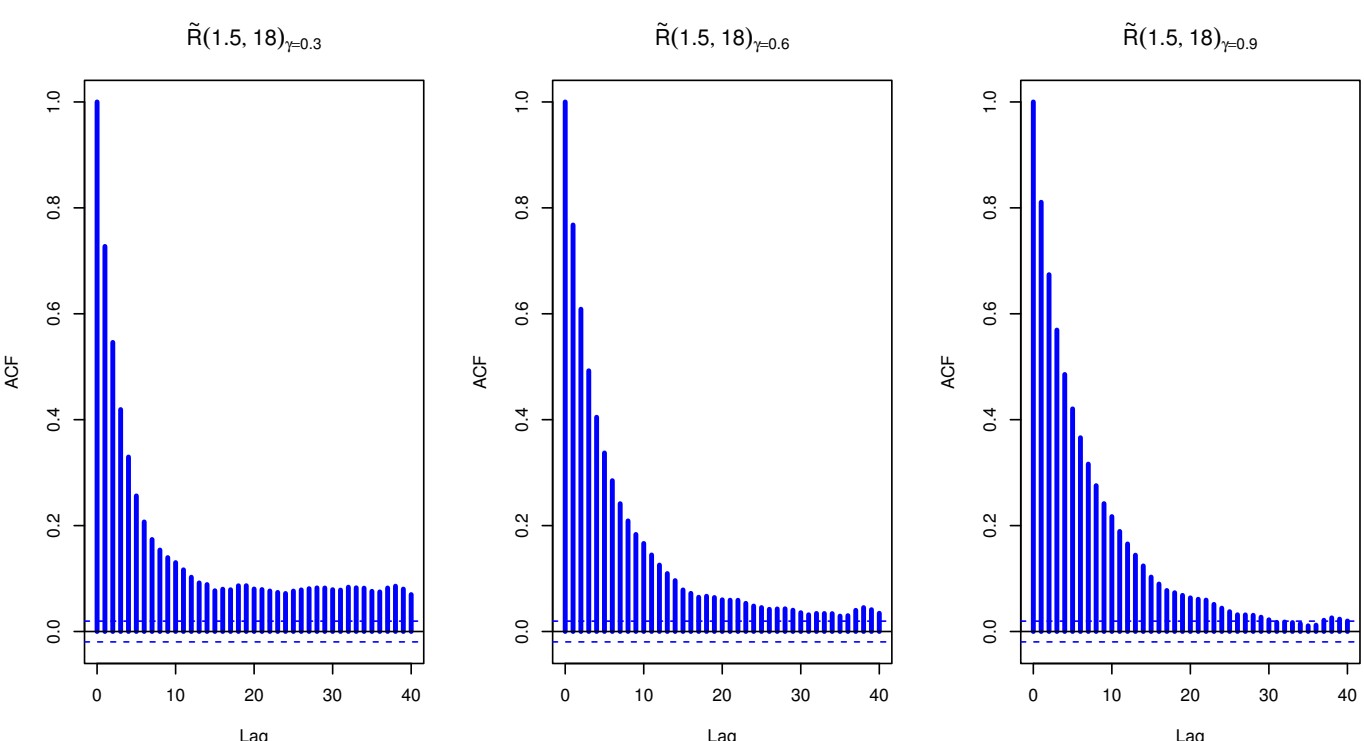

**Figure 6.** The auto-correlation plot of MCMC for fuzzy reliability $\tilde{R}$ of TIIHLW distribution where lower is 1.5 and upper is 18.

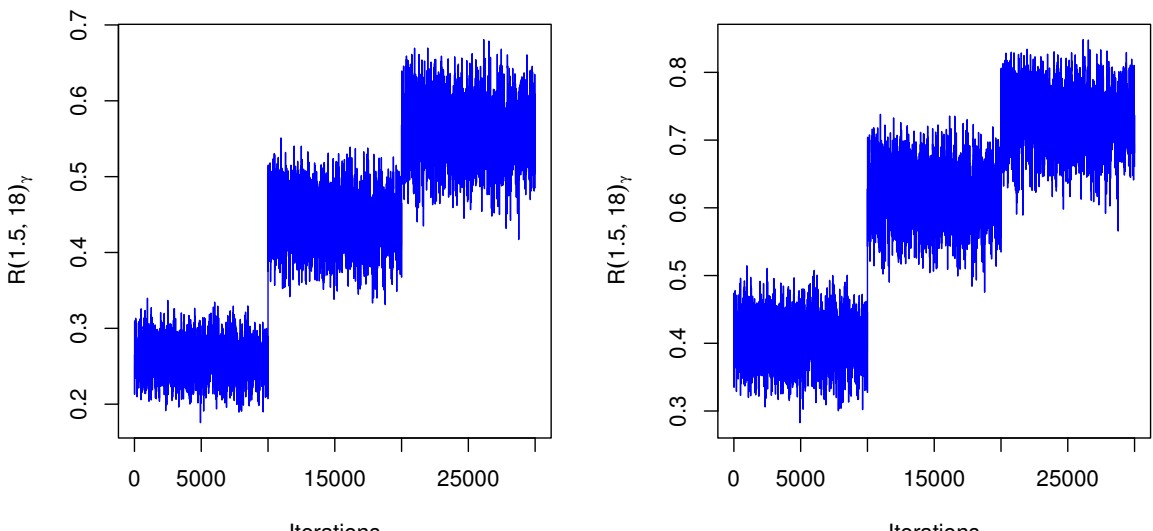

**Figure 7.** The auto-correlation of fuzzy reliability $\tilde{R}$ by MCMC results of TIIHLW distribution.

## 8. Conclusions

In this paper, we introduced classical and Bayesian estimation approaches for fuzzy reliability estimation using the lifetime Type II Half Logistic Weibull distribution model, using it as a base. We calculated parameters and reliability fuzzy estimations using MLE and Bayesian procedures. In addition, the asymptotic confidence intervals were created. The MCMC method is used to create Bayesian credible intervals. We use Monte Carlo simulation to compare the results of different methods. The simulation results indicate that, in the case of point estimation, the performances of the classical and Bayesian estimators are nearly identical, especially for large sample sizes. The results of the simulation study indicated that the MSEs of Bayes estimates for informative priors were significantly lower than those of the others. In addition, the HPD credible intervals outperform the competition. The simulation results indicate that reliability under fuzzy is better than traditional reliability for all sample sizes, and that fuzzy reliability at Bayes estimates is better than the maximum likelihood method. The empirical study using a real data set for bladder cancer patients to evaluate the flexibility of the TIIHLW distribution fits well with the proposed distribution.

**Author Contributions:** Conceptualization, R.A.H.M. and D.A.R.; methodology, R.A.H.M., A.H.T., E.M.A. and D.A.R.; software, E.M.A.; validation, A.H.T., R.A.H.M. and D.A.R.; formal analysis, D.A.R. and E.M.A.; investigation, R.A.H.M. and A.H.T.; resources, R.A.H.M., A.H.T. and D.A.R.; data curation, R.A.H.M. and D.A.R.; writing—original draft preparation, A.H.T., E.M.A. and D.A.R.; writing—review and editing, A.H.T., R.A.H.M., E.M.A. and D.A.R.. All authors have read and agreed to the published version of the manuscript.

**Funding:** This research received no external funding.

**Data Availability Statement:** Data are available in this paper.

**Acknowledgments:** All thanks and appreciation to the editorial board and reviewers.

**Conflicts of Interest:** The authors declare no conflict of interest.

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
