# Peer review of "Inference of Reliability Analysis for Type II Half Logistic Weibull Distribution with Application of Bladder Cancer"

_axioms, doi:10.3390/axioms11080386_

Round 1

Reviewer 1 Report

This research deals with two approaches for fuzzy reliability estimation using lifetime Type II Half Logistic Weibull distribution model. Paper is interesting and can be published after minor revision with following remarks.
1. What is the reason for an analysis of the maximum likelihood and Bayes methods as estimation methods for the present research? Other wide-spread methods should be described. What are advantages and disadvantages of the considered approaches?
2. Introduction part should be improved with more detailed analysis of published results of other authors.
3. It should be better if authors could analyze an influence of each parameter (alpha, beta, lambda) presented in Fig. 1 on the shape and behavior of the probability density and hazard rate function.
4. The used scenarios for the parameters (alpha, beta, lambda) should be explained.
5. Conclusion part should be improved with important results.
6. There are some typos within the text.

Author Response

We start  by  thanking the reviewers for their careful reading of our paper and for helping us to improve its quality and presentation.

We have carefully read the reviewer’s comments on our previous version of the manuscript, and we have rewritten the paper with their suggestions in mind. The modifications in the manuscript are highlighted in blue color, and the answers in this reply letter are also marked in blue.

We give below the specifics of our response to the reviewers .

Reviewer 2 Report

In the manuscript , the authors introduce classical and Bayesian estimation approaches for fuzzy reliability estimation using lifetime Type II Half Logistic Weibull distribution . They calculated parameters and reliability fuzzy estimations using MLE and Bayesian procedures, also creating the asymptotic confidence intervals. The MCMC method is used to create Bayesian credible intervals.

 The paper is correctly written and presents a good methodological approach. There is an adequate Conclusions section.  The presentation should perhaps made more clear for  non-expert in  fuzzy reliability, as outined below. Apart this, it is well presented, organized and structured.

The topic is important or interesting enough to both scholars and reliability engineers.  There are enough references, and   the paper is reasonably self-contained.

Decision: The manuscript is appropriate for the journal in terms of both scope and quality.  The reviewer deems it is acceptable as it is , with the minor suggestions that follow:

1) Since  fuzzy reliability theory is generally much les known than "classical"  or traditional  reliability theory, the authors should make an effort, for the non expert readers,  spending more time and space, to give a better introduction to fuzzy reliability theory and explaining with more practical examples its advantages over  traditional reliability theory. For instance, the difference between the fuzzy reliability and the traditional  reliability computations appear not clear (for the reviewer) when reading sec. 5 (Bayesian Estimation), appearing “hidden” in eqns from (5.4) to (5.9), without any comment.

In synthesis, the only statement (as far as I can see) on the issue is insufficient, i.e.the one appearing from the 2nd line f the Introduction , i.e.: “It is well established that traditional probability-based reliability analysis is insuffcient for dealing with failure data and modelling uncertainty. To address this issue, the concept of a fuzzy approach has been used to the evaluation of a system's reliability.

2) At page 6, there is perhaps a wrong or not well explained statement, concerning the confidence interval estimator, in the framework of classical  statistic (so it seems, since the MLE is discussed). The statement:

“The probability that interval includes the parameter value is what we call the confidence level”.

is not correct in  in the framework of classical  statistics, or should be better explained since , if w is the the parameter value and (a,b) is a  confidence interval , the  probability P(a < w<b) is not defined (or is a trivial quantity, i.e. 0 or 1), being (a,b,w) NOT random variables.

The statement, instead, has meaning in the the framework of Bayesian statistics, since W is a random parameter so that P(a < W<b) is well defined.

3) The following phrases should be corrected "Since the ML estimators of the parameters cannot be defined in analytic forms. Therefore, the actual distributions of ML estimators cannot be" derived.

(perhaps the full stop between forms and Therefore should be  corrected into a comma). 

4) page 13) the following statement is unclear: "The simulation results indicate that fuzziness is better than actual for all sample sizes.."

The adjective actual perhaps lacks a substantive (fuzziness is a a substantive, it cannot be compared to an adjective, as far as I understand). Moreover, is the term " actual "defined in the paper? . In the Introduction, at page 1, the term "traditional reliability" is used, so I would expect a comparison between  "fuzzy reliability theory" and "traditional  reliability theory" , if this is what the authors mean (else, what they mean  should be better explained).

Author Response

(The authors gave the same response as above.)
